# Study protocol for a randomised controlled trial of a virtual antenatal intervention for improved diet and iron intake in Kapilvastu district, Nepal: VALID

Naomi M Saville ,[1] Sanju Bhattarai,[2] Helen Harris-Fry,[3] Santosh Giri,[2] Shraddha Manandhar,[2] Joanna Morrison,[1] Andrew Copas [ID],[1] Bibhu Thapaliya,[2] Abriti Arjyal,[2] Hassan Haghparast-Bidgoli [ID],[1] Sushil C Baral,[2,4] Sara Hillman[5]

[1]Institute for Global Health, University College London, London, UK
[2]HERD International, Kathmandu, Nepal
[3]Department of Population Health, London School of Hygiene and Tropical Medicine, London, UK
[4]HERD, Kathmandu, Nepal
[5]Institute for Women's Health, University College London, London, UK

**Correspondence to**
Dr Naomi M Saville;
n.saville@ucl.ac.uk

## ABSTRACT

**Introduction** Despite evidence that iron and folic acid (IFA) supplements can improve anaemia in pregnant women, uptake in Nepal is suboptimal. We hypothesised that providing virtual counselling twice in mid-pregnancy, would increase compliance to IFA tablets during the COVID-19 pandemic compared with antenatal care (ANC alone.

**Methods and analysis** This non-blinded individually randomised controlled trial in the plains of Nepal has two study arms: (1) control: routine ANC; and (2) 'Virtual' antenatal counselling plus routine ANC. Pregnant women are eligible to enrol if they are married, aged 13–49 years, able to respond to questions, 12–28 weeks' gestation, and plan to reside in Nepal for the next 5 weeks. The intervention comprises two virtual counselling sessions facilitated by auxiliary nurse midwives at least 2 weeks apart in mid-pregnancy. Virtual counselling uses a dialogical problem-solving approach with pregnant women and their families. We randomised 150 pregnant women to each arm, stratifying by primigravida/multigravida and IFA consumption at baseline, providing 80% power to detect a 15% absolute difference in primary outcome assuming 67% prevalence in control arm and 10% loss-to-follow-up. Outcomes are measured 49–70 days after enrolment, or up to delivery otherwise. Primary outcome: consumption of IFA on at least 80% of the previous 14 days. Secondary outcomes: dietary diversity, consumption of intervention-promoted foods, practicing ways to enhance bioavailability and knowledge of iron-rich foods. Our mixed-methods process evaluation explores acceptability, fidelity, feasibility, coverage (equity and reach), sustainability and pathways to impact. We estimate costs and cost-effectiveness of the intervention from a provider perspective. Primary analysis is by intention-to-treat, using logistic regression.

**Ethics and dissemination** We obtained ethical approval from Nepal Health Research Council (570/2021) and UCL ethics committee (14301/001). We will disseminate findings in peer-reviewed journal articles and by engaging policymakers in Nepal.

**Trial registration number** ISRCTN17842200.

## STRENGTHS AND LIMITATIONS OF THIS STUDY

⇒ The virtual counselling intervention under test with pregnant women is novel and tailored to a pandemic situation where opportunities for face-to-face antenatal counselling may be limited.

⇒ The intervention design to engage pregnant women and their family members in a problem-solving interactive discussion is based on formative research on the barriers to uptake of iron and folic acid (IFA) supplements and antenatal care and our process evaluation analysis of context will help us to understand the external validity of the results.

⇒ Carefully designed electronic data collection forms enable thorough follow-up of participants, drawing on their information at recruitment where needed to validate outcome data.

⇒ Necessary adaptation to the COVID-19 context led to delays in starting the trial, constraining the time frame within the funding period. This means that the time frame to enrol each woman, provide her with two sessions of virtual counselling, and capture her outcomes is shorter than may have been ideal.

⇒ Our study carries some risk of bias. As with all non-blinded trials we could not eliminate the potential for selection bias. Our self-reported outcomes also carry risk of social desirability and recall bias. The electronic questionnaires included internal consistency and range checks to validate responses for each outcome, and to minimise recall bias on the primary outcome (IFA consumption) the interviewers triangulated responses using a series of probes which asked how many days the participant consumed IFA, and how many days were missed, before confirming the final number with the respondent.

## INTRODUCTION

Anaemia in pregnancy is a complex public health problem that leads to severe adverse maternal and child health outcomes, including maternal morbidity and mortality[1–4]

and impaired infant growth.[5] No country in South Asia is on course to meet the World Health Assembly target of a 50% reduction in anaemia by 2025[6] and estimates from Nepal's 2016 Demographic and Health Survey indicate that 46% of pregnant women were anaemic (haemoglobin concentration <110 g/L)[7]—almost no change from the 2011 estimate of 48%.[8]

Despite the lack of progress in reducing anaemia in pregnancy, and its multifactorial aetiology,[9–11] there are several effective interventions that could help. Systematic reviews of randomised controlled trials show that consumption of iron and folic acid (IFA) supplements and deworming with antihelminthics in pregnancy can both lower the risk of maternal anaemia—by 70% and 15%, respectively.[12 13] There are additional health benefits of IFA supplementation beyond anaemia, including increased birth weight,[14–16] child survival[17] and development.[18]

Provision of free-of-cost IFA and antihelminthics have been scaled up in Nepal from 20 weeks' gestation as part of routine care in Nepal. The challenge lies in supporting women to access and use these evidence-based interventions.[19] Current access to recommended antenatal interventions is improving but remains suboptimal. In 2016, 41% of women took iron tablets on at least 180 days of pregnancy, 69% took antihelminthics and the same proportion had four antenatal care (ANC) clinic visits.[8]

Although these aspects of ANC have improved markedly between 2011 and 2016,[7 8] the COVID-19 pandemic threatens to compromise this progress.[20] The pandemic has seen a sharp rise in neonatal and maternal mortality, along with declines in institutional deliveries and quality of care.[21 22] In Kapilvastu district of Nepal during the early phase of the pandemic in 2019/2020, Maternal Mortality Ratio increased from 0.0 to 14.4 per 100 000 live births. Between 2018 and 2020, having at least one ANC visit decreased from 84% to 74% and having four ANC visits decreased from 63% to 52%, which would likely decrease uptake and adherence to IFA.[23–26] Repeated lockdowns, concerns about going to a health facility, discrimination against frontline workers (Female Community Health Volunteers, FCHVs) and loss of livelihoods could all pose barriers to women's access to ANC and IFA.[22]

Nutrition education and counselling interventions have had variable effects on maternal anaemia.[27] In Nepal, the complexity of factors limiting access to adequate nutrition and IFA require gender-sensitive, tailored, participatory counselling methods, rather than a 'one-size-fits-all' didactic educational approach.[28–30] A recent trial of a social norms intervention in India found large effects on consumption of IFA consumption, suggesting that such an approach holds promise.[31]

A question remains as to whether and how these types of interventions can be adapted to reach the most geographically and socially isolated, who have least access to antenatal and social support. Virtual interventions may offer a solution, with almost all (93%) of Nepalese households now owning a mobile phone.[7] Systematic reviews have shown that 'eHealth' and 'mHealth' can improve a range of nutrition outcomes, including physical activity and healthy eating behaviour,[32] but research has been concentrated in high-income contexts. Use of telemedicine for maternal and newborn care during the pandemic has been beneficial, but difficulties in reaching high-risk groups and maintaining quality of maternity services remain in low-income countries.[33] In rural Nepal, virtual counselling has the potential to address the shortage of healthcare professionals, and transportation and travel time barriers to accessing healthcare.[34]

## Study aims and hypotheses

In our study, designed in accordance with the Standard Protocol Items: Recommendations for Interventional Trials guidelines,[35] we test whether a virtual counselling intervention offered to pregnant women living in Kapilvastu could offer a feasible, affordable, scalable and equitable solution to increase consumption of IFA supplements.

The primary objective of the Virtual Antenatal Intervention for Improved Diet and iron intake (VALID) trial is to assess whether providing pregnant women with an antenatal virtual counselling intervention, in addition to usual government services, increases women's compliance to IFA supplementation, compared with pregnant women who only had access to usual government services. Compliance is defined as IFA consumed on 12 out of the last 14 days before the endline interview (which is 49–70 days after baseline). We hypothesised that the antenatal virtual counselling intervention would increase pregnant women's compliance to IFA tablets.

Secondary objectives are as follows:
1. To assess whether providing an antenatal virtual counselling intervention to pregnant women improves
   - Dietary diversity,
   - Timing and number of antenatal visits (ANC),
   - Consumption of recommended iron-rich foods
   - Practice of iron absorption-enhancing behaviours
   - Knowledge of iron-rich foods and anaemia risks in pregnancy.
   - Knowledge of COVID-19 symptoms, prevention and vulnerable population groups
2. To evaluate the feasibility, acceptability, equity, cost, cost-effectiveness and reach of the antenatal virtual counselling intervention (using an Implementation Science approach).

## METHODS AND ANALYSIS
## VALID trial design and setting

This is a non-blinded two-arm individually randomised controlled trial, with an allocation ratio of 1:1, conducted in Kapilvastu district, Lumbini Province of Nepal.

Trial arms are:
1. Control: access to routine ANC.
2. 'Virtual counselling': comprising of two interactive virtual counselling sessions provided to pregnant women

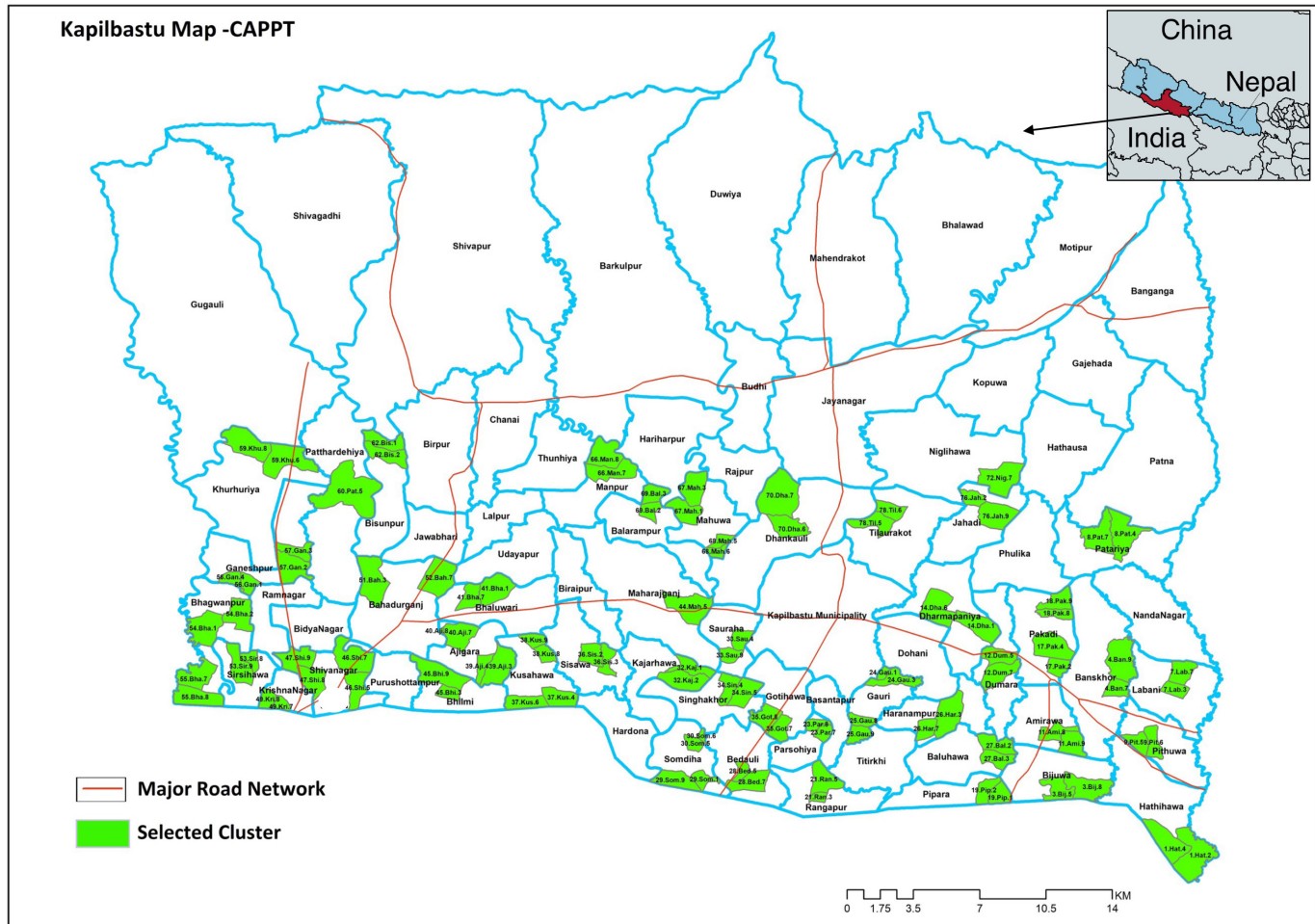

**Figure 1** Map of study clusters.

in mid-pregnancy, using 'zoom' on a tablet, in addition to routine ANC.

The trial is being implemented in 54 clusters in Kapilvastu district, Lumbini Province of Nepal, in the Western Terai (plains) of Nepal, bordering Uttar Pradesh state of India. The district population is 569 844 and literacy rates of women and men are 45% and 65%, respectively.[36] Hinduism is the main religion, followed by Islam and the predominant ethnicity is Madhesi, many of whom speak Awadhi language. Anaemia prevalence is high (45%) and IFA in pregnancy is suboptimal: 43% took at least the recommended dose of 180 IFA tablets, 33% took 60–179 tablets and 24% took <60 tablets.[7]

### Eligibility

Study clusters (in figure 1) which rural areas of southern Kapilvastu district that do not adjoin the main East-West highway that traverses Nepal; with no major market; 1100–3199 projected population; surrounded with a buffer zone of non-study clusters; and >50% Madhesi (plains ethnicity) as per the pretrial census.

Pregnant women are eligible for a baseline survey if they are aged 13–49 years, able to respond to questions, reside in a study cluster and consent to participate. Additional inclusion criteria for enrolment in the trial include: 12–28 weeks' gestation (estimated from recall of last menstrual period or expected date of delivery given by a health worker), does not plan to leave the country during the 5 weeks since enrolment and no other pregnant woman in her household already enrolled in the trial.

To enrol participants, interviewers identify pregnant women in their study areas with help from FCHVs. They confirm eligibility, take written consent and then phone the office to receive the random allocation.

### Randomisation

A stratified block randomisation process is used to allocate pregnant women enrolled into the trial intervention or control arm. To balance the trial arms by two strong preidentified determinants of IFA consumption, parity and baseline IFA consumption status, randomisation is done within each of four strata: (IFA yes/first pregnancy); (IFA yes/ not first pregnancy); (IFA no/ first pregnancy) and (IFA no/ not first pregnancy). For each stratum a separate allocation sequence is prepared. To make the sequence unpredictable, random permutation of the allocations within blocks is conducted and block sizes vary randomly between sizes 8, 6 and 4 using 'blockrand' package in R programming software. Then, sequential allocations are sealed into sequentially

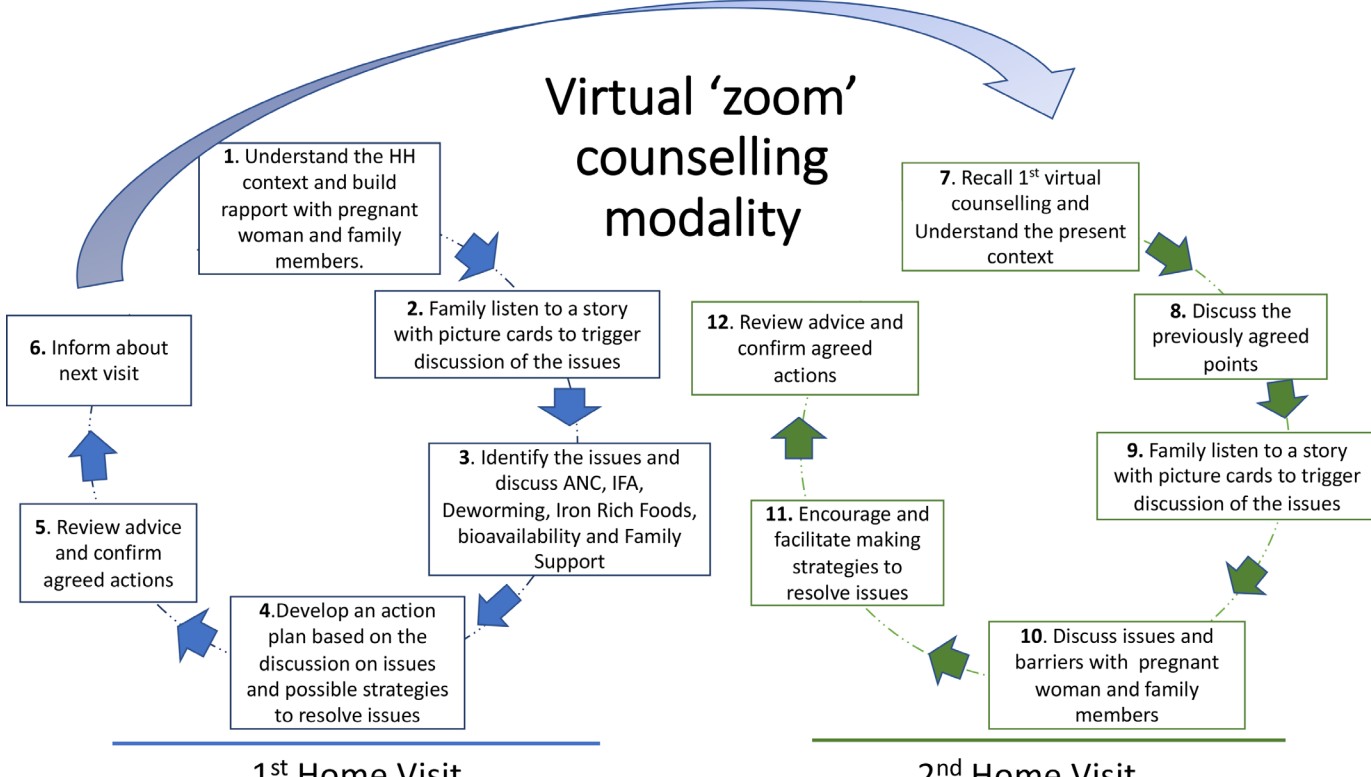

**Figure 2** Virtual counselling intervention model. ANC, antenatal clinic; IFA, iron and folic acid; HH, household.

numbered opaque envelopes by the project coordinator in Kathmandu, transported to the field office, stored in a locked cabinet and opened by the monitoring and evaluation manager when the enumerators call to find out a pregnant woman's allocation at the end of the baseline interview. After assignment of pregnant women to study arms, blinding of trial staff and participants is impossible, since interviewers distribute of tablet devices to pregnant women in the intervention arm.

### Intervention
After enrolment, interviewers call pregnant women in the intervention arm to schedule appointments to deliver the tablets and orientate women on how to use the tablets. Tracking of tablet delivery is made by entering this information onto CommCare. Each tablet holds a sim card and has a data package which is topped up as necessary.

Enrolled women are assigned to 1 of 10 counsellors, who are recruited and trained auxiliary nurse midwives. Counsellors call pregnant women, using case lists provided in the CommCare app on their tablets, to schedule their tailored virtual counselling sessions via Zoom. Pregnant women are offered their first counselling session after enrolment, at 12–28 weeks' gestation, and the second counselling session is around 2 weeks later, at 14–31 weeks' gestation. A diagram summarising the virtual counselling intervention is shown in figure 2.

The counselling aims to support women and their families to take actions to change dietary practices, take IFA and deworming tablets and access ANC. Counselling is conducted with pregnant women and their families

to support them to think critically about the causes of anaemia in pregnancy in their household and community, using stories and inductive questioning to trigger dialogue and reflection.[37] Stories directly address issues identified from formative research.[38] At each session, pregnant women (and their families if they attend) are engaged in a cycle of action and reflection whereby they discuss common issues that affect women in pregnancy and examples of actions for improving intake of IFA and deworming tablets, accessing ANC and dietary practices. At the end of the first counselling session, the pregnant women and families make specific action plans to address the issues that are relevant for their family. In the second counselling session, these action plans are reviewed and further discussed to support women and their families to address the issues and a second action plan is made. The tablets are collected back by interviewers after completing two counselling sessions.

The auxiliary nurse midwives received in-depth training on how to engage families in problem solving, the dialogical approach and how to use a reference manual with examples of actions that pregnant women and their families could take. The training also provided counsellors with knowledge on anaemia consequences and prevention, and logistical training on how to use the tablets and record their progress.

Adherence to intervention protocols is ensured by monitoring the counselling (by supervisors joining remotely), by video-recording sessions, and by requiring counsellors to provide copies of action plans and to fill

**Table 1** VALID outcome measures

| VALID trial outcomes | Recall period | Definition | Variable type | Effect measure to compare arms/summary statistic |
|---|---|---|---|---|
| **Primary outcome** | | | | |
| Proportion of pregnant women complying with recommended IFA tablet intake | 14 days | IFA consumed on 12 or more days out of the previous 14 days (ie, on at least 80% of days) | Binary | Difference in proportion |
| **Secondary outcomes** | | | | |
| Dietary diversity | 24 hours | Count of the number of food groups consumed in the previous 24 hours preceding the endline interview, assessed using the list-based method, out of 10 food groups, as defined by (ref) | Count (0 to 10) | Difference in mean |
| Consumption of intervention-promoted foods | 24 hours | Any consumption of green leafy vegetables, meat or fish | Binary | Difference in proportion |
| Practicing one or more to enhance bioavailability | 7 days | Recalled one or more of the following: using lemon or other vitamin C-rich foods with meals, eating sprouted grains or pulses, avoiding tea/coffee 1 hour either side of meals, or spreading meat-eating over two eating occasions rather than one. | Binary | Difference in proportion |
| Knowledge of iron-rich foods | N/A | Count of iron-rich food groups correctly recalled | Count (0 to 9) | Difference in mean |
| ANC visits | Baseline to endline | Count of antenatal (ANC) visits between enrolment and endline interview | Count (0 to 6) | Difference in mean |
| **Exploratory outcomes** | | | | |
| Understanding of why blood tests are taken at antenatal check-ups. | N/A | Proportion of those women who had a blood test at ANC who could correctly explain one or more reason for having a blood test. | Binary | Difference in proportion |
| Knowledge of COVID-19 symptoms, precautions and vulnerability | N/A | Proportion of women who could correctly identify at least one vulnerable group, at least three COVID-19 control/prevention measures and at least three COVID-19 symptoms | Binary | Difference in proportion |
| ANC visits at the right time for her gestational age | Pregnancy to date | Whether the woman had her ANC visits at 2, 4, 6 and 8 months or not | | |

ANC, antenatal clinic; IFA, iron and folic acid; VALID, Virtual Antenatal Intervention for Improved Diet and iron intake.

paper and electronic forms about each counselling session. Counsellors manage workload by prioritising women approaching 28 weeks' gestation and filling a 'phone call log' in CommCare to record outcome of each call.

Participants are free to seek concomitant care during pregnancy irrespective of their trial allocation.

### Outcomes

The primary outcome of the trial is the proportion of pregnant women consuming IFA on at least 80% of the previous 2 weeks (ie, on ≥12 out of 14 days recalled). Other outcomes are given in table 1.

### Outcome measurement

Interviewers measure outcomes on trial participants at enrolment (12–28 weeks' gestation) and endline (ideally 49–70 days after baseline, otherwise up to delivery). All outcomes are recalled by trial participants during interviews and recorded in electronic questionnaires. Additionally, at baseline we collect the woman's age, parity, medical history, date of the last menstrual period, pregnancy symptoms/problems, and other key socioeconomic and demographic information.

The data collection tools are programmed onto mobile devices in Nepali and English on Android operating system tablets or mobile phones using the CommCare electronic data collection platform.[39] These have in-built jump-sequences and value limits to prevent entry of data outside plausible ranges. As interviewers are fluent Awadhi speakers but do not read in Awadhi, they conduct the interviews by reading the question in Nepali and wording it into Awadhi to the respondent. Standard Awadhi wording is agreed on within the team. Interviewers follow standard procedures for 24-hour dietary recall measurements and IFA recall to ensure that measurements are accurate and inter-observer difference minimised.

The monitoring team meets monthly and Kathmandu-based team members provide support as needed. Interviewers log any problems with electronic forms and the data team make corrections reversibly in the data using Stata data cleaning 'do' files as required.

### Sample size

Our target sample size is 300 pregnant women, 150 in each arm. With at least 270 followed up (90% follow-up rate in each arm), we have 80% power to detect a 15 percentage-point increase in IFA compliance (the proportion of women consuming IFA on at least 12 out 14 days preceding the endline survey), if the control arm

compliance prevalence is 66%–68% (or 16.7 percentage point difference if control arm compliance is only 50%).

To achieve the required sample size, interviewers communicate with the Female Community Health Volunteers (FCHVs) in the study area. In line with government norms, FCHVs are provided with an incentive per day spent identifying pregnant women for our trial. To increase the response rate in later pregnancy, we provide an incentive (200 Nepali rupees mobile phone top-up) to trial participants after the endline interview. Target dates for enrolment and endline interviews, based on gestational age and date of enrolment, are displayed on customised lists in CommCare on interviewers' mobile devices.

## Statistical analysis

The trial statistician (AC) and Nepal principal investigator (NS) will make a statistical analysis plan, to detail the analysis strategies, covariate adjustment and the approach to any missing data. We will present this to the Trial Steering Committee and Data Monitoring Committee for approval before analysis. Primary analysis will be replicated by a second analyst.

Analyses will involve graphically presenting the number of days of IFA consumption in last 14 days by arm (bar chart) at baseline and follow-up by trial arm. Primary analysis will be by intention-to-treat using two-sided testing at the standard 5% significance level. Analysis of our binary primary outcome, 12+ days consumption of IFA in last 14 days at follow-up, is based on logistic regression adjusting for baseline consumption in the previous 14 days, gestational age at interview and parity (note the randomisation strata are based on parity and baseline consumption). Our 'fully adjusted' results (secondary approach) will include additional adjustment for wealth score constructed out of household assets, education status of pregnant woman and age.

We will follow a similar methodology for the analysis of secondary and exploratory outcomes (given in table 1), and process indicators to be reported in a separate publication. We will use linear regression for continuous outcomes, logistic regression for binary outcomes and negative binomial regression for 'count' outcomes such as number of ANC visit between baseline and endline. We will examine the distribution of continuous and count outcomes before analysis of intervention effectiveness.

We plan to also report the intervention effect stratified by consumption of IFA at baseline, and test for an interaction with intervention arm. Within the intervention arm we will report and test the association between the degree of intervention exposure (number of counselling sessions received) and IFA consumption at endline. Other subgroup analyses will compare the intervention effect on the primary outcome by socioeconomic status (wealth quintile), education and exposure to ANC.

While our main analyses are under the intention-to-treat principle, for the primary outcome we will also conduct a 'per protocol' analysis in which only women who received virtual counselling are compared with all participants in the control arm.

We expect very low missing data in our baseline indicators, but we anticipate some women will not have a primary outcome. We consider data as missing where the woman is unavailable, has moved away or withdrew consent but we do not regard data as 'missing' if due to miscarriage. We are not planning imputation of the primary outcome since there is little information on which to base the imputation other than baseline IFA consumption, which we are including as a covariate in our regression models.

## Data management

We use CommCare's case management system that enables interviewers and counsellors to follow-up pregnant women for data collection and intervention and promotes participant retention. Interviewers and FCHVs hold paper lists of potential pregnant women prior to enrolment. Data collected on electronic forms are locally stored before they are synchronised (encrypted) to the CommCare cloud server.

The data manager in Kathmandu downloads the data daily from the CommCare cloud server onto their server, imports the data into Stata and runs 'do' files for data pseudonymising, labelling and recoding. The data are stored on password-protected, encrypted, secure server computers in lockable rooms at the HERD International Kathmandu office. We back up data from physical servers onto secure cloud servers daily, and onto external hardware-encrypted hard drives each week. Person-identifiable data are stored in separate encrypted files and are only used to generate follow-up lists for authorised field team members. All other data are pseudonymised. Study arm is encoded but not labelled. After data collection is completed, we will delete VALID data from laptops, tablets, and the CommCare server.

For qualitative data collected in Awadhi and Nepali languages, qualitative interviewers transcribe the data in Nepali. They then send audio recordings and transcriptions to the qualitative data officer who checks completeness, anonymises and stores the data in a lockable cabinet.

Data cleaning is completed by identifying outliers and removing as required. HERD International follows international and Nepal Health Research Council (NHRC) policies for archiving of data in Nepal. We will make the pseudonymised dataset open access by using the University College London (UCL) data sharing platform as per the UK Medical Research Council (MRC) guidelines. Any request for archived person-identifiable data will go through the trial management team in alignment with provisions in the consent forms.

No storable biological specimens are collected during this study.

## Process evaluation

We are conducting a mixed-methods process evaluation to describe the feasibility, acceptability, equity, reach and

| Intervention | Pregnant woman and her household members Dialogue and problem solving in two virtual 'zoom' counselling session on a tablet | | | | | |
|---|---|---|---|---|---|---|
| Awareness and health literacy (household members and pregnant woman) | Awareness of iron-rich food sources | Awareness of ways to improve bioavailability | Awareness of the need for iron-rich diet & iron and folic acid tablets (IFA) | Awareness of ways to reduce side-effects of iron and folic acid (IFA) | Awareness of role of HH in improving diets, increasing access to antenatal care and supporting IFA consumption | |
| Household actions to support Pregnant Woman | Encourage healthy snacking | Cooking and serving practices to increase Pregnant Women's access to iron-rich diet | Encourage / enable dietary practices to increase bioavailability of iron | Encourage / enable practices to minimise side-effects of IFA | Encourage / enable consumption of IFA | Accompany to ANC & retrieve re-supply of IFA |
| Enabling environment | Increased household support to Pregnant Woman | | Men take a more active role in supporting Pregnant Woman | | IFA and 'eating well' not linked with having a large baby | |
| Intermediate outcomes | Improved intake of iron-rich foods | Improved dietary diversity | Improved actions to increase iron absorption | Improved uptake of Antenatal Care | Improved access and uptake of IFA | Improved access to and uptake of de-worming |
| Primary / secondary outcomes | Improved compliance in taking Iron and Folic Acid tablets Dietary modifications to increase iron consumption / absorption in Pregnant Woman | | | | | |
| 'Higher level' outcomes not measured | Improved dietary iron adequacy | Improved absorption of iron from diet | Improved overall dietary adequacy | Improved micronutrient status | Higher haemoglobin / Less anaemia | Improved physical health of mother and baby |

**Figure 3** Theory of change for the Virtual Antenatal Intervention for Improved Diet and iron intake intervention. IFA, iron and folic acid.

fidelity to plans, as guided by the theory of change[38 40] in figure 3.

Quantitative process outcomes are listed in table 2.

We will use qualitative data (semistructured interviews) to explore how the context may influence intervention effect.[38] We will analyse post-counselling forms and observe virtual counselling sessions to describe intervention implementation and fidelity to plans. Further, we will conduct focus group discussions with supervisors and counsellors to explore how context affects the intervention and analyse factors affecting the families' interaction during the counselling sessions. We will interview 20 pregnant women who received the intervention, purposively sampled based on the number of children they have and their socioeconomic status. We will also interview 15 family members of different pregnant women (husbands and mothers-in-law) who did and did not participate in the intervention to explore the factors affecting their participation and their perceptions of the intervention. To consider the sustainability of the intervention we will show a piloted virtual counselling session to health workers in two health facilities and discuss the feasibility and acceptability of using virtual counselling within the health system and community. In addition, we will interview two FCHVs to explore the factors affecting recruitment and enrolment of women and their observation on others' (especially participants') perceptions of the intervention.

## Cost effectiveness

We will estimate cost and cost-effectiveness of the intervention from a provider perspective by estimating the costs of designing and implementing the virtual counselling (programme costs) and costs to public healthcare providers. We collect programme cost data from HERD international project accounts, staff time use surveys and interviews with project staff. We estimate costs to public healthcare providers resulting from any increased demand for health services caused by the intervention, using ANC and IFA data from the endline interview and any available secondary data on unit costs for these services. If statistically significant results are found, we will calculate incremental cost-effectiveness of virtual counselling compared with routine care and estimate incremental cost-effectiveness ratios for the primary outcome (ie, proportion of pregnant women complying with recommended IFA intake) and the secondary outcome Women's Dietary Diversity Score. We will estimate ICER in terms of the cost per number of women who complied with recommended IFA intake and the cost per unit change in dietary diversity. If possible, we will estimate disability adjusted life years (DALYs) based on the

**Table 2** Quantitative process evaluation outcomes

| Domain | Measure | Data source |
|---|---|---|
| Acceptability | Proportion of pregnant women who:<br>▶ were satisfied with virtual counselling session<br>▶ consented to participate in the trial out of those eligible at baseline<br>▶ had family member(s) attending the virtual counselling session<br>▶ reported the tablet and video link is an acceptable way to receive counselling and advice during pregnancy | Endline interview |
| Effectiveness | ▶ received satisfactory support from family members<br>▶ implemented the action plan<br>▶ found the intervention useful<br>▶ understood what was said during the virtual counselling session<br>▶ needed support to use the tablet and video link<br>▶ reported constant internet disruption for smooth conversation during virtual counselling session<br>▶ felt confident in their nutrition knowledge and/or able to take actions to improve nutrition/health | Endline interview |
| Feasibility | ▶ completed two virtual counselling sessions<br>▶ had direct access to a phone/device<br>▶ were easily able to use the tablet and video link<br>Number of attempts to contact participant when intervention not possible | Endline Interview<br>Phone call log |
| Fidelity | Proportion of counsellors adhering to the virtual counselling manual guidance | Observations of sessions |
| Coverage and reach | Proportion of<br>▶ pregnant women enrolled out of those approached<br>▶ marginalised Muslim and Dalit castes women enrolled out of those approached<br>▶ primiparous women enrolled of those approached<br>▶ young (<18 years of age) enrolled of those approached | Baseline interview |
| Sustainability | Proportion of organisations (government and non-governmental) attending the dissemination workshop willing to implement the intervention | Dissemination workshop |
| Cost | Total intervention cost from programme provider perspective<br>Total intervention costs by line items/inputs<br>Cost of Intervention per pregnant woman | Costing tools and time allocation sheets |

primary outcome and present the results in terms of cost per DALYs averted. A series of one-way sensitivity analyses will be conducted to assess impact of uncertainties on the cost-effectiveness results.

### Trial monitoring and safety

We do not expect virtual counselling to have negative effects, although we have an established complaints procedure. Asking women to discuss their pregnancy may be difficult or embarrassing; participants can stop an interview or counselling at any time. Counsellors are trained to advise on how to mitigate minor IFA side effects, such as constipation and indigestion, or conflict in the household from provision of tablets.[2 41] If the counsellor or interviewer observes any pregnancy danger signs or the woman is feeling unwell at the time of interaction, they advise the woman to seek care from the nearest health facility, as per the government's standard treatment protocol.[42]

The virtual counselling intervention is designed to limit the spread of COVID-19, but interviewers need to visit participants' home to provide the tablets. We removed measurements requiring physical contact (anthropometry and haemoglobin) from our protocol to enable interviewers to keep 2 m from household members, except while demonstrating use of the tablet. Field staff are required to adhere to COVID-19 Infection Prevention and Control protocols.

Counsellors and interviewers record adverse events (including maternal deaths or COVID-19 cases) and inform field managers who submit an adverse events form to the Principal Investigators . As the trial follow-up period is short, we will not have interim audits. We do not expect to end the counselling early (unless the COVID-19 pandemic renders activities impossible), and we do not envisage ancillary or post-trial care will be needed beyond routine ANC.

### Ethics and dissemination

We have obtained ethical approval from NHRC (approval number: 570/2021) with amendment approval on 9 March 2022, UCL ethics committee (Project ID number: 14301/001 with amendment approval on 6 May 2022). We will seek approval for any major changes to the protocol from the Trial Steering Committee and ethics boards and amend the registry.

General consent was sought from municipality leaders, health system managers and ward representatives. Interviewers seek household heads' oral consent to speak with pregnant women, then take written consent/assent from pregnant women. Additional consent is obtained from guardians if the participant is aged <18 years. After assessing eligibility, the interviewers take consent to enrol eligible pregnant women into the trial by signature or thumbprint. In subsequent visits to pregnant women, interviewers and counsellors take oral consent. Information sheets and consent forms in English are available in online supplemental Annex 1 (copies in Nepali and Awadhi are available on request).

Our consent forms ask permission to share the anonymised data with other researchers for secondary analyses and to revisit the participant for follow-up. The final data

| TIMEPOINT** | STUDY PERIOD | | | | | |
| --- | --- | --- | --- | --- | --- | --- |
| | Enrolment | Allocation | Post-allocation | | | Close-out |
| | $t_0$ Baseline interview | | $t_1$ ($t_0$ + 1 to 7 days or more#) | $t_2$ ($t_1$ + 1 to 7 days or more#) | $t_3$ ($t_2$ + 14 days or more#) | $t_x$ *Endline interview* ($t_0$ + 49 to 70 days ideally but permitted up to delivery#) |
| **ENROLMENT at baseline interview:** | | | | | | |
| Eligibility screen for baseline | X | | | | | |
| Informed consent for baseline | X | | | | | |
| Eligibility screen for trial | X | | | | | |
| Informed consent for trial | X | | | | | |
| **Allocation** | | X | | | | |
| **INTERVENTIONS:** | | | | | | |
| *Tablet handover and orientation* | | | X | | | |
| *Virtual counselling session 1* | | | | X | | |
| *Virtual counselling session 2* | | | | | X | |
| **ASSESSMENTS:** | | | | | | |
| *Baseline survey variables:* Socioeconomic indicators / food security; Marriage / Pregnancy history; Uptake / quality of antenatal care (ANC); Iron and Folic Acid (IFA) access/consumption; Deworming access/consumption; Dietary Diversity Score; Knowledge and intake of key iron-rich foods; Practice of iron-absorption enhancement | X | | | | | |
| *Process variables* **Post-counselling form** Ease of conducting counselling; Who attended counselling?; Pregnant woman's level of comfort; Pregnant woman's attentiveness; Engagement of family members; Actions agreed upon with family | | | | X | X | |
| **plus in Post-2nd counselling session only** Extent actions were implemented; Factors preventing / facilitating taking actions | | | | | X | |
| *Endline interview variables:* Uptake / quality of antenatal care (ANC); Iron and Folic Acid (IFA) access/consumption; Deworming access/consumption; Dietary Diversity Score; Knowledge and intake of key iron-rich foods; Practice of iron-absorption enhancement; Confidence in health knowledge; Exposure and response to intervention; Awareness of COVID-19 prevention, vulnerability and symptoms | | | | | | X |

**Figure 4** Schedule of enrolment, interventions and assessments. Interviewers enrolled women from 14 January until 23 February 2022. Endline interviews began on 9 March 2022 and are due to be completed by 19 June 2022, depending on the availability of participants. # Note that although target follow-up periods were defined at outset, lack availability of respondents to attend counselling and/or interviews means that gaps between time points are varied.

will be held by UCL and HERD International on secure servers. The analysis dataset and statistical code will be shared in a web annex to the main trial publication.

### Patient and public involvement

We conducted qualitative formative research with reproductive age women, mothers-in-law, fathers and FCHVs to inform the design of the study, using findings to refine our research questions/priorities and shape the intervention.[29] We did not involve patients/public in the choice of outcomes but involved FCHVs and health workers in participant recruitment. When disseminating study results, we will engage trial participants and municipal leaders from the study clusters as well as national and provincial policymakers, government officials, academics and other relevant stakeholders. We will prepare a policy brief in English and Nepali and distribute it at dissemination workshops.

We will report results to NHRC within 1 year of the end of the study, publish our results in peer-reviewed journals and present findings at national, regional and international conferences where possible.

### Current trial status

The trial timeline is provided in figure 4.

**Acknowledgements** We thank Suprich Sapkota, Basudev Bhattarai, Avinash Bhurtel, Anjali Basnet and Samata Yadav for their contributions in development of this protocol.

**Contributors** NS led the development of the protocol and the writing of the manuscript draft. SB gave oversight to field management, all trial processes and drafted the process evaluation and randomisation processes. SG oversaw data collection tools with NS. JM, SM, AA and BT designed the intervention and process evaluation. AC, NS, HH-F and SG designed the trial methodology, sample size calculation, data analysis plan and outcome measurement. HH-B designed the economic evaluation. SB, NS, SCB and SH provided overall leadership. All authors provided inputs and read and approved the final paper.

**Funding** This work was supported by UK Medical Research Council (MRC)/ Newton Fund, grant number MR/R020485/1. The funding body has no role in the study design, collection, analysis, or interpretation of data, in the writing of the manuscript or the decision to submit the report for publication.

**Map disclaimer** The inclusion of any map (including the depiction of any boundaries therein), or of any geographic or locational reference, does not imply the expression of any opinion whatsoever on the part of BMJ concerning the legal status of any country, territory, jurisdiction or area or of its authorities. Any such expression remains solely that of the relevant source and is not endorsed by BMJ. Maps are provided without any warranty of any kind, either express or implied.

**Competing interests** None declared.

**Patient and public involvement** Patients and/or the public were involved in the design, or conduct, or reporting or dissemination plans of this research. Refer to the Methods section for further details.

**Patient consent for publication** Not applicable.

**Provenance and peer review** Not commissioned; externally peer reviewed.

**ORCID iDs**
Naomi M Saville http://orcid.org/0000-0002-1735-3684
Andrew Copas http://orcid.org/0000-0001-8968-5963
Hassan Haghparast-Bidgoli http://orcid.org/0000-0001-6365-2944

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
