## [Reviewer comments · BMJ Open]

ARTICLE DETAILS

TITLE (PROVISIONAL)	A study protocol for a randomised controlled trial of a Virtual Antenatal Intervention for improved Diet and Iron intake in Kapilvastu district, Nepal: VALID
AUTHORS	Saville, Naomi; Bhattarai, Sanju; Harris-Fry, Helen; Giri, Santosh; Manandhar, Shraddha; Morrison, Joanna; Copas, Andrew; Thapaliya, Bibhu; Arjyal, Abriti; Haghparast-Bidgoli, Hassan; Baral, Sushil; Hillman, Sara

VERSION 1 – REVIEW

REVIEWER	Fernandes Nilson, Eduardo Augusto Ministry of Health of Brazil, Department of Health Promotion
REVIEW RETURNED	08-Aug-2022

GENERAL COMMENTS	This study aims to evaluate a virtual counselling intervention offered to pregnant in terms of feasibility, affordability, scalability and equity to increase consumption of iron-folic acid supplements. The protocol is very innovative, complete and detailed and is likely to provide interesting inputs to micronutrient deficiency policies, especially those aimed at remote communities. Some brief comments: Regarding the strengths and limitations, are there further limitations that could be added? For example, the scalability of the intervention and extrapolation of the results to other contexts? In terms of the cost-effectiveness analysis, more details could be provided about the evaluated outcomes (most details are on cost components).
---

REVIEWER	Hegerty, Christopher Queensland Health, Warwick Hospital
REVIEW RETURNED	08-Nov-2022

GENERAL COMMENTS	The study appears to have been completed five months ago so it's a bit puzzling why the protocol is being peer reviewed now. This may be a question more for the journal than the authors. It is a pity the 'harder' end points of Hb level and birthweight were unable to be used rather than the fairly soft 'reported use of IFA tablets' but the authors have explained the reasons. I think the the wording on the information sheet for participants at the bottom of page 32 under 'Expected outcome of the Trial' is unfortunate. By using the words 'understand how' rather than 'understand whether' this seems to be predicting the outcome and telling the participants what the trialists expect/want to find, and this seems to me a problem in an unblinded trial with the outcome based on participants answers to a questionnaire, particularly
---

	when the participants know which arm of the trial they are in and are being rewarded for participation. This should probably be discussed in the 'Strengths and limitations' section, as possibly should other issues relevant to this type of trial to do with possible selection, recall and confirmation biases.
--	--

VERSION 1 – AUTHOR RESPONSE

Responses to Reviewer Reports:

Reviewer: 1

Dr. Eduardo Augusto Fernandes Nilson, Ministry of Health of Brazil, University of Sao Paulo

Comment: This study aims to evaluate a virtual counselling intervention offered to pregnant in terms of feasibility, affordability, scalability and equity to increase consumption of iron-folic acid supplements. The protocol is very innovative, complete and detailed and is likely to provide interesting inputs to micronutrient deficiency policies, especially those aimed at remote communities.

Response: Thank you for the positive feedback.

Comment: Regarding the strengths and limitations, are there further limitations that could be added? For example, the scalability of the intervention and extrapolation of the results to other contexts?

Response: We have added to the second bullet point in the strengths and limitation section saying:

“...and our process evaluation analysis of context will help us to understand the external validity of the results.” We have also added a point in the limitations on risk of bias (see below).

Unfortunately, there was not sufficient space in the limitations section to add a bullet point on feasibility of scale-up, so we have not stated that further tests of scale-up may be needed.

Comment: In terms of the cost-effectiveness analysis, more details could be provided about the evaluated outcomes (most details are on cost components).

Response: we have added further details on the outcomes to be included in the cost-effectiveness analysis at the end of the cost-effectiveness section.

Reviewer: 2

Dr. Christopher Hegerty, Queensland Health, Queensland Government Department of Health and Ageing

Comments to the Author:

Comment: The study appears to have been completed five months ago so it's a bit puzzling why the protocol is being peer reviewed now. This may be a question more for the journal than the authors.

Response: We agree that the review process took a long time but are grateful to have finally received feedback.

Comment: It is a pity the 'harder' end points of Hb level and birthweight were unable to be used rather than the fairly soft 'reported use of IFA tablets' but the authors have explained the reasons.

Response: We agree entirely that it would have been far better to have been able to measure haemoglobin (Hb) levels, but unfortunately the Government of Nepal was not permitting measurements requiring physical contact at the time of applying for ethical approval. We had funding to undertake a different trial, in which Hb was an outcome, but this had to be halted due to the COVID-19 pandemic restrictions in Nepal. Our virtual counselling intervention was designed as a minimum contact intervention to replace a face-to-face counselling intervention.

Comment: I think the wording on the information sheet for participants at the bottom of page 32 under 'Expected outcome of the Trial' is unfortunate. By using the words 'understand how' rather than 'understand whether' this seems to be predicting the outcome and telling the participants what the

trialists expect/want to find, and this seems to me a problem in an unblinded trial with the outcome based on participants answers to a questionnaire, particularly when the participants know which arm of the trial they are in and are being rewarded for participation.

Response: We thank the reviewer for picking up this mistake in the English version of the consent form. We have checked the wording in the Nepali and Awadhi participant information sheets, which contain the exact wordings read out to participants (none of the respondents had the English version read to them). The wording in Nepali and Awadhi is as follows (important wording shaded yellow):

Nepali:

यस अध्ययन अन्तर्गत गर्भवती महिलालाई ट्याबलेट मार्फत भर्चुअल तरिका प्रयोग गरेर गरिएको पोषण परामर्शले गर्भवती महिलाले गर्भावस्थामा चाहिने आइरन र फोलिक एसिडको नियमित सेवन, आइरन युक्त खाने कुराहरू खान र गर्भ जाँचमा के कस्तो सुधार ल्याउन सकिन्छ भन्ने कुरा बुझ्न मद्दत गर्नेछ।

Awadhi:

यी अध्ययनमे गर्भवती महतारी के ट्याबलेटसे भर्चुअल तरिका प्रयोग कइ के करल पोषण परामर्शसे गर्भवती महतारी के गर्भावस्थामे चाहेवाला आइरन औ फोलिक एसिडके नियमित सेवन, आइरन होअल खाना खायेक अउर गर्भ जाँचमे केसन सुधार के सकाजात है कहेवाला बतिया बुझेक मद्दत होइ ।

When we back translated this, we came up with the following alternative translation:

“In this study, the virtual nutritional counselling given to pregnant women through a tablet will help us to understand what kinds of improvements can be made in the regular intake of iron and folic acid, the consumption of iron-rich foods, and antenatal check-ups during pregnancy. As nutritional assistants will encourage antenatal check-ups and institutional delivery during the virtual counselling sessions, we hope that antenatal check-ups and institutional delivery will increase.”

We have replaced the translation in the English participant information sheet with this text and uploaded it with our revisions. The Nepali and Awadhi say “help us to understand what kind of improvements can be made” rather than “help understand how the virtual nutrition counselling intervention can improve”. Although this is a subtle difference, we think that the message that we expect the trial to have a positive impact comes across less strongly in the Nepali and Awadhi.

Also, within the same participant information sheet we say the following:

“Are there any benefits if you participate?

There is no direct benefit of participating in this study, however you might enjoy interacting with the counsellor on the tablet. If found effective and feasible, the results of this trial will provide evidence that counselling can be provided virtually to improve health and nutrition of pregnant women from excluded communities in Kapilbastu and rest of Nepal.”

We believe therefore that despite the mistake in not saying “whether the trial has impact” the general message in the information sheet is that the trial is trying to generate evidence and not prove that the intervention works. For this reason, we have not added a statement in the limitations section.

Comment: This should probably be discussed in the 'Strengths and limitations' section, as possibly should other issues relevant to this type of trial to do with possible selection, recall and confirmation biases.

Response: Thank you for this suggestion. We have added a point about the fact that we could not eliminate all sources of bias:

“Our study carries some risk of bias. As with all non-blinded trials we could not eliminate the potential for selection bias. Our self-reported outcomes also carry risk of social desirability and recall bias. The electronic questionnaires included internal consistency and range checks to validate responses for each outcome, and to minimize recall bias on the primary outcome (IFA consumption) the interviewers triangulated responses using a series of probes which asked how many days the participant consumed IFA, and how many days were missed, before confirming the final number with the respondent.”